# Reproductive Cessation and Post-Reproductive Lifespan in Honeybee Workers

**DOI:** 10.3390/biology13050287

**Published:** 2024-04-24

**Authors:** Karolina Kuszewska, Anna Woloszczuk, Michal Woyciechowski

**Affiliations:** 1Department of Zoology and Animal Welfare, Faculty of Animal Science, University of Agriculture in Krakow, Al. Mickiewicza 21, 31-120 Krakow, Poland; 2Institute of Environmental Sciences, Jagiellonian University, 30-387 Krakow, Poland

**Keywords:** *Apis mellifera*, honeybee, menopause, expected longevity, post-reproductive lifespan

## Abstract

**Simple Summary:**

Explanations for the evolution of menopause in humans and other animals concern the indirect fitness benefits that older females can receive by caring for the offspring of their children and grandchildren. Knowledge about the post-reproductive lifespan is still incomplete and until recently was documented only in nonhuman primates, a few species of toothed whales, and guppies, though it has also been documented in insects for gall-forming aphids and parthenogenetic ants. In our paper, we provide evidence that menopause also exists in honeybee societies. Honeybee workers are usually sterile. Nevertheless, the absence of a queen and her pheromones in a nest stimulates workers to activate their ovaries and lay unfertilized eggs that develop into males. However, even in queenless colonies, some workers do not activate their ovaries. In this study, we show that physiologically older workers have less activated ovaries than physiologically younger individuals, indicating that the possibility of workers’ reproduction decreases with workers’ physiological age. This may happen because bees require a minimum of two weeks to activate their ovaries, meaning physiologically older workers can die before producing their own eggs. Thus, investing energy in helping relatives care for their offspring rather than channelling that energy into one’s own reproduction can positively influence workers’ fitness.

**Abstract:**

The post-reproductive lifespan is an evolutionary enigma because the cessation of reproduction in animals seems contrary to the maximization of Darwinian fitness. Several theories aim to explain the evolution of menopause, one of which suggests that females of a certain age receive more fitness benefits via indirect selection (kin selection) than they would directly from continuing reproduction. Post-reproductive lifespans are not very common in nature but have been described in humans, nonhuman primates, a few species of toothed whales, guppies, and in some insect societies consisting of clonal colony members, such as aphid and ant societies. Here, we provide evidence that menopause also exists in honeybee societies. Our study shows that workers with a short life expectancy (older and/or injured workers) invest fewer resources and less time in their own reproduction than workers with a long life expectancy (younger and/or uninjured workers), even if their colony is hopelessly queenless. These results are consistent with the kin selection explanation for the evolution of menopause and help us understand the net effects of relatedness and social cooperation in animals.

## 1. Introduction

Explanations for the evolution of the post-reproductive lifespan in humans and other animals concern the indirect fitness benefits that older females can receive by caring for the offspring of their children and grandchildren [1]. Evidence for this adaptive significance of the post-reproductive lifespan in humans derives from a study based on demographic records from the 18th and 19th centuries in Finland and Canada, where the reproductive success of individuals was shown to be strongly associated with the grandmother’s longevity [2]. However, the post-reproductive lifespan phenomenon is not limited to human females but has also been reliably documented in a wide range of female vertebrates, including nonhuman primates [3], a few species of toothed whales [4], and guppies [5]. Moreover, post-reproductive lifespans have also been documented in insects, specifically, in the gall-forming aphid *Quadrartus yoshinomiyai*, where post-reproductive females self-sacrifice to defend the colony and reproductive individuals [6], and in the parthenogenetic ant *Pristomyrmex punctatus*, where workers switch their roles from reproduction to foraging as they age [7]. In all of the above examples except guppies, the animals live in kin groups and care for their relatives or their relatives’ offspring, which probably affords them more indirect fitness benefits than would continuing their own reproduction.

Eusocial insects, such as ants and honeybees, provide examples of extreme altruism based on kin selection [8]. The workers generally behave altruistically by restricting their own reproduction or refraining from it altogether [9]. In the honeybee (*Apis mellifera*), workers are usually sterile in the presence of a queen, because her pheromones inhibit the development of ovaries in workers [10]. When the queen and her pheromones are absent from the colony, honeybee workers can activate their ovaries and lay unfertilized eggs that develop into males (drones) [10]. When colonies permanently lose their queen (i.e., are ‘hopelessly queenless’), workers’ reproduction becomes significant. Hopelessly queenless colonies most likely arise following the death of the queen on her mating flight, which naturally occurs with a probability of 14–35% [11,12]. The drones reared in queenless nests are viable and have some chance of participating in mating [13], whereas, for the workers, the period following the queen’s death is a final opportunity for them to reproduce before the colony perishes owing to a lack of replacement workers [14,15]. Researchers have suggested that orphaned workers should focus on their own reproduction and stop performing demanding and risky tasks such as foraging [16]. However, previous studies have shown that queenless honeybee workers engage in both personal reproduction and other colony tasks, including brood food production, foraging trips, and colony defense tasks [17]. Nevertheless, some workers in queenless colonies remain sterile and do not activate their ovaries, which is usually explained by genotypic variation in workers from different patrilines [18] or the environmental conditions surrounding worker development [19]. These components (genetic and environmental) can influence the reproductive potential of workers by altering the number of ovarioles in the ovary, which is strongly correlated with ovary activation and personal reproduction [20]. Another possible explanation for the presence of workers with inactivated ovaries in queenless colonies involves the physiological age of these workers at the time the colony is orphaned. Physiologically older workers can be expected to avoid activating their ovaries because such activation is metabolically costly [21] and workers have no guarantee of survival until the time of laying their own eggs. Therefore, one can expect older workers to invest their resources and time in post-reproductive altruism.

The physiological age of bees, which can be measured as life expectancy (the average period that an individual may expect to live), is affected not only by chronological age but also by other factors, such as genetic factors [22], environmental conditions [23], disease, and injury [24,25]. Unhealthy or injured individuals have shorter expected longevities than healthy and uninjured individuals [24,26] and can be expected to be physiologically older than healthy and uninjured bees of the same age. In the present study, we tested whether physiologically older workers cease personal reproduction. We compared ovarian development among workers from different age cohorts (15, 18, 21, 24, 27, 30, 33, and 36 days old) and among workers from the same age cohorts but with a shortened life expectancy due to injury imposed by puncturing the last segment of the thorax with a needle (for details, see the Methods and Figure 1). We expected injured workers to show less ovarian development than uninjured workers of the same chronological age, and we expected older workers to show less ovarian development than younger individuals within the same group (control or injured).

## 2. Methods

This research was performed in July and August 2010 in the experimental apiary in Krakow (southern Poland). Three queenright honeybee (*A. m. carnica*) colonies were studied, with each colony consisting of 20,000–40,000 workers. All colonies were treated the same way, following the experimental designs of previous papers [26,27,28]. First, frames with newly emerged bees were moved from each colony (day 1) to an incubator (36 °C). All workers that emerged within 24 h were divided into two groups: (1) an untreated control group and (2) a group injured by puncturing the last segment of the thorax with a needle (diameter: 0.35 mm; puncture depth: to the first drop of hemolymph) to shorten their life expectancy [26,29]. The workers from both groups were marked on the thorax with a spot of paint that differed in color between the groups (Marabu Brilliant Painter) and were returned to the native colony as soon as possible. This procedure was repeated eight times for every colony, and bees were marked every time with colors that varied based on treatment, colony, and the day when the cohorts were marked. The number of individuals in each colony and group is provided in Appendix A. Every colony included bees from the two groups (control and injured) of each age class (Figure 1). When the last group of workers was marked and returned to the hive (21st day of the experiment), each colony was temporarily orphaned by removing the queen. Each queen was transferred to a new hive box with a small group of workers. As a result of this manipulation, experimental workers from each group (control and injured) were orphaned at different ages (1, 4, 7, 10, 13, 16, 19, and 22 days old). Two weeks after the colonies were orphaned (day 36), the queens were returned to their native colonies, and all marked workers from both groups (control and injured) were recaptured. This time (two weeks) was needed for orphaned workers to begin ovary activation [30,31]. We counted the number of surviving workers of each cohort and treatment group and then killed the bees by freezing (the number and percentage of surviving recaptured bees are presented in Appendix A). Workers from every group, colony, and age category (a maximum of 20) were dissected under a stereomicroscope (binocular loupe), and the number of ovarioles, extent of ovarian development, and size of the hypopharyngeal gland (HPG) were examined. HPG size was calculated from the average size of 10 acini, which are sac-like dilations comprising the compound HPG (square root of longest acinus x shortest diameters of 5 right-gland and 5 left-gland acini; Figure 2A). The total number of ovarioles in both ovaries of each worker was recorded, and the ovarian development of all dissected bees was assessed. To characterize ovarian development, the most developed ovariole of each ovary was selected, and the maximum diameter of the two ovarioles (the maximum width) was measured as described by Nakaoka [32] (Figure 2B,C). Additionally, we assessed ovarian development on a relative scale described previously [27,33]: 1, non-activated ovary; 2, previtellogenic activated ovary; 3, vitellogenic ovary with developing oocytes; and 4, mature ovary with at least one egg. This scale helped reveal the distribution of ovarian development in every colony, age category, and treatment group (the results and associated figures are provided in the Appendix A). 

Differences in the survival of workers between the control and experimental (injured) groups were analyzed using factorial linear regression. The measurements included the percentage of surviving bees in every colony, with the age of bees and the experimental group (control and injured) as fixed factors. Similarly, ovariole number, ovarian development, and HPG size were analyzed using factorial linear regression, with the experimental group (control and injured) and bee age as fixed factors. For this purpose, the average of these parameters (ovariole number, ovarian development, and HPG size) was calculated previously for every colony, age category, and experimental group. If the results of the analyses revealed an interaction between age and experimental group, then the factorial linear regression was followed by a simple linear regression for every experimental group. All calculations were performed with Statistica 13.3.

## 3. Results and Discussion

First, to test whether workers from older cohorts and workers from injured groups varied in life expectancy, we measured the percentage of individuals that survived to a specific age in each of the control and injured groups. The probability of survival was negatively correlated with worker age and varied with experimental treatment (factorial linear regression: *p* < 0.006, β = −0.78, R^2^ = 0.665, t_45_ = −2.869). The control individuals had a longer life expectancy than the injured workers (*p* = 0.003; β = −1.01, R^2^ = 0.665, t_45_ = −3.050). However, no interaction was found between worker age and treatment (*p* = 0.270, β = 0.467, R^2^ = 0.665, t_45_ = 1.117; Figure 3A), indicating that expected longevity decreased with age in a similar manner in both the control and injured groups. These results are in accordance with the results of previous studies showing that the life expectancy of workers is affected not only by age but also by factors such as disease [34], poisoning, and injury [24,26]. The shorter life expectancy of sick, poisoned, or injured bees and their older physiological age can be associated with the activation of oxidative processes [35]. There is evidence that various external abiotic (extreme temperatures, dehydration, environmental pollutants, and radiation) and biotic (pathogen attack) stress factors induce the excessive production of reactive oxygen species (ROS), causing an imbalance between oxidative stress and antioxidative defense mechanisms [35,36,37,38,39]. This situation leads to damage of DNA, lipids, and proteins and thus to numerous physiological violations, cell death, and, finally, the entire organism dying [35,36,40]. 

Having determined that the workers in different age and treatment groups indeed had different lifespans, we next examined whether they also differed in anatomical parameters. First, we assessed the development of the HPGs, which synthesize and store brood food [41]. Young bees generally have large HPGs with high rates of protein synthesis [42], whereas older foragers have smaller, less active glands [42]. Consistent with previous studies, our results revealed that HPG size was negatively correlated with worker age (*p* < 0.001, β = −0.61, R^2^ = 0.692, t_45_ = −5.657; Figure 3B). As we expected, injury negatively impacted the size of the HPG (*p* < 0.001, β = −1.61, R^2^ = 0.692, t_45_ = −5.516), most likely because the injured workers were physiologically older than the control workers of the same age. The interaction between factors (age and treatment) was also statistically significant, indicating that the change in HPG size with age varied between treatments (*p* < 0.001, β = −1.61, R^2^ = 0.692, t_45_ = −5.516). Simple regressions performed separately for the control and injured groups of workers of different ages showed that in the control group, the glands became smaller with worker age (*p* < 0.001, β = −0.72, R^2^ = 0.523, t_45_ = −4.916; Figure 2B), whereas in the injured group, no correlation was identified between gland size and worker age (*p* = 0.124, β = −0.32, R^2^ = 0.104, t_45_ = −1.600; Figure 3B). The lack of a correlation between age and gland size in the injured group is not surprising. The youngest workers analyzed in this study were 15 days old, and all of them may have been foragers, which have small and degenerated HPGs [43]. In typical honeybee colonies, workers usually begin foraging at 18–28 days of adult life [10]; however, workers with shorter longevities, such as our injured bees, are known to start foraging at an earlier age than control workers with normal life expectancies [24], which influences their HPG development.

To test our hypothesis that physiologically older workers do not activate their ovaries because such activation is metabolically costly, we assessed the number of ovarioles and ovary activation in workers subjected to different treatments (injury and control) and in workers of different ages. The number of ovarioles in the ovary is determined during the larval period and depends on the quality and quantity of food [10] and the presence of a queen and her mandibular gland pheromones during larval development [27,44]. The number of ovarioles is usually stable in adult honeybee workers and declines only in rare cases [45]. In contrast, ovary activation is determined in the adult life of workers and depends not only on the presence of a queen and her pheromones, the presence of a brood, and the tasks undertaken by the workers [10,33] but also on the number of ovarioles in the ovary, among other factors. Previous studies showed that the number of ovarioles is positively correlated with ovary activation in honeybee workers [20]. Therefore, we investigated whether our experimental workers exhibited differences among one another in ovariole number. The results showed that workers from different age classes (*p* = 0.121, β = 0.31, R^2^ = 0.125, t_45_ = 1.582; Figure 3C) and treatment groups (*p* = 0.789, β = −0.15, R^2^ = 0.125, t_45_ = −0.269; Figure 3C) did not differ in the number of ovarioles and that the interaction between age and treatment was nonsignificant (*p* = 0.955, β = −0.03, R^2^ = 0.125, t_45_ = −0.056; Figure 3C). However, ovary activation was negatively correlated with worker age (*p* < 0.001, β = −0.79, R^2^ = 0.474, t_45_ = −5.464; Figure 3D), and injured workers had fewer activated ovaries than control individuals of the same age (*p* = 0.001, β = −1.33, R^2^ = 0.474, t_45_ = −3.37). Moreover, the interaction between factors (age and treatment) was statistically significant (*p* = 0.020, β = 0.98, R^2^ = 2.402, t_45_ = 2.40); thus, simple regressions were performed separately for the control and injured groups of workers of different ages. In both the control (*p* < 0.001, β = −0.67, R^2^ = 2.402, t_45_ = −4.285; Figure 3D) and injured groups, ovarian development was negatively related to worker age (*p* = 0.002, β = −0.58, R^2^ = 2.402, t_45_ = −3.378; Figure 3D), indicating that the possibility of worker reproduction decreased with worker age or, more precisely, with worker life expectancy. Bees require a minimum of two weeks under favorable conditions (e.g., queenless conditions) to activate their ovaries and start laying eggs [10,46]. Therefore, compared with workers with a higher life expectancy, workers with a shorter life expectancy have a higher probability of dying before their ovaries start to produce eggs. It is possible that such physiologically older individuals invest their energy in helping relatives care for offspring rather than investing it in their own reproduction. This interpretation is in accordance with predictions of the grandmother hypothesis, which explains the evolution of the post-reproductive lifespan in humans and other animals. We acknowledge that, in our study, we did not evaluate how the physiologically older workers allocated their energy to various tasks in the nest. However, previous papers showed that some tasks in the nest, such as foraging, are performed by older individuals, even in hopelessly queenless colonies [47,48]. 

The phenomenon of the post-reproductive lifespan in insects is not limited to honeybees and has been described in the clonal gall-forming aphid *Q. yoshinomiyai*, where post-reproductive females self-sacrifice to defend the colony and reproductive individuals [6]. However, in aphids, colony members are clones, and kin selection conflicts do not exist, which has promoted the evolution of post-reproductive altruism [6]. Another example is the parthenogenetic ant *P. punctatus* [49], where colony members switch their roles from reproduction to foraging and assisting their sisters in reproduction as they age. It is possible that similar to the ants described above, honeybee workers that are physiologically older do not invest energy or resources in their own reproduction but instead invest them in other tasks in the nest to help their full and half-sisters [10]. The observed cessation of reproductive behavior and the investment of resources to help relatives in honeybees are in accordance with the predictions of kin selection [8,9]. In addition, the results of our investigation are consistent with the grandmother hypothesis [1,2], which attempts to explain the evolution of post-reproductive lifespans in animals and suggests that within kin groups, menopause enhances fitness by producing post-reproductive individuals (originally grandmothers, as the hypothesis was first proposed to explain the evolution of menopause in humans) who can assist their reproductive relatives (originally adult daughters). Honeybee workers that do not have sufficient time to activate their ovaries enhance their fitness by assisting other colony members (full sisters and half-sisters).

In conclusion, we have shown that honeybee workers with a short life expectancy do not invest their resources in reproduction, even if the colony is hopelessly queenless. These results are in accordance with kin selection theory and help us understand the net effects of relatedness and social cooperation in animals.

## 4. Conclusions

In short, we have found that the ovarian development of honeybee workers depends on their physiological age. Individuals with a shorter life expectancy, such as older and/or injured individuals, do not activate their ovaries. As such, we propose that honeybee workers exhibit a kind of menopause.

## Figures and Tables

**Figure 1 biology-13-00287-f001:**
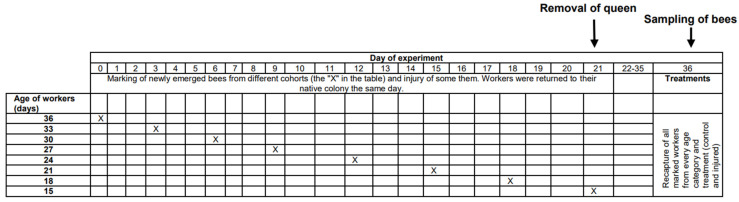
Timeline of the experiment showing the manipulations on particular days. Details are presented in the Methods section.

**Figure 2 biology-13-00287-f002:**
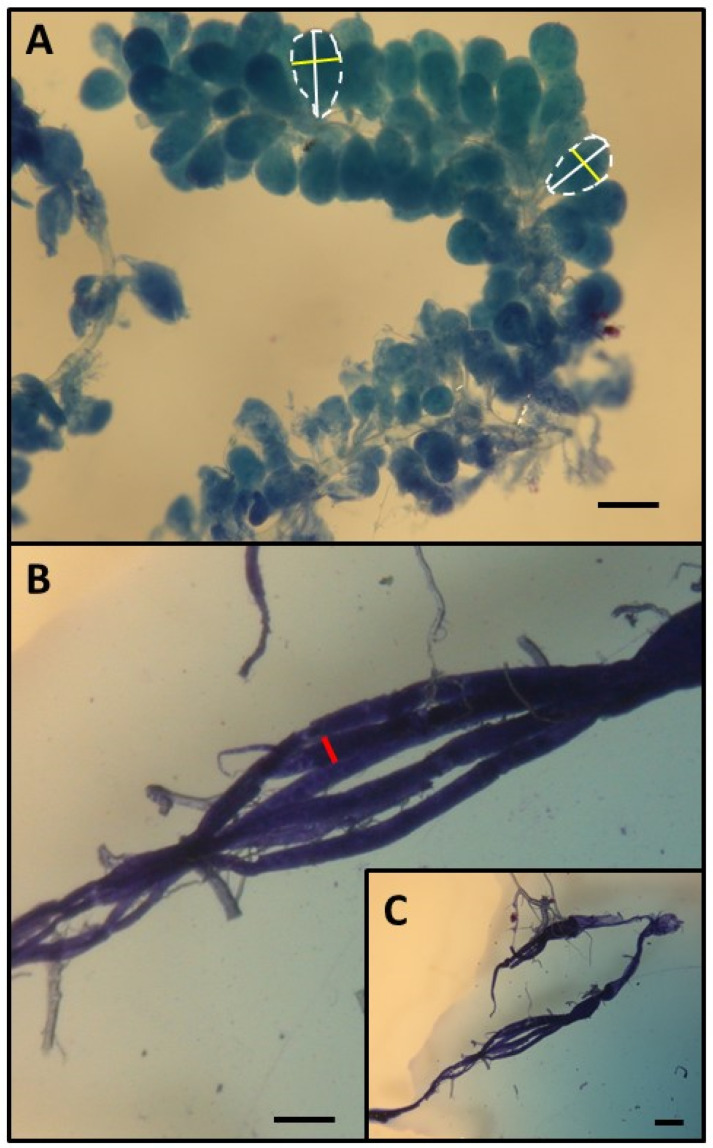
Determination of HPG (**A**) and ovary size (**B**,**C**). HPG size was calculated from the average size of 10 acini (the square root of longest acinus x the shortest diameters of 5 right-gland and 5 left-gland acini). Each single acinus is outlined with a broken line and the longer diameters are indicated by a white bar while the shorter diameters are indicated by a yellow bar. To characterize ovarian development, the most developed ovariole of each ovary was selected, and the maximum diameter (red bar) of the two ovarioles (the maximum width). Scale bars indicate 100 µm.

**Figure 3 biology-13-00287-f003:**
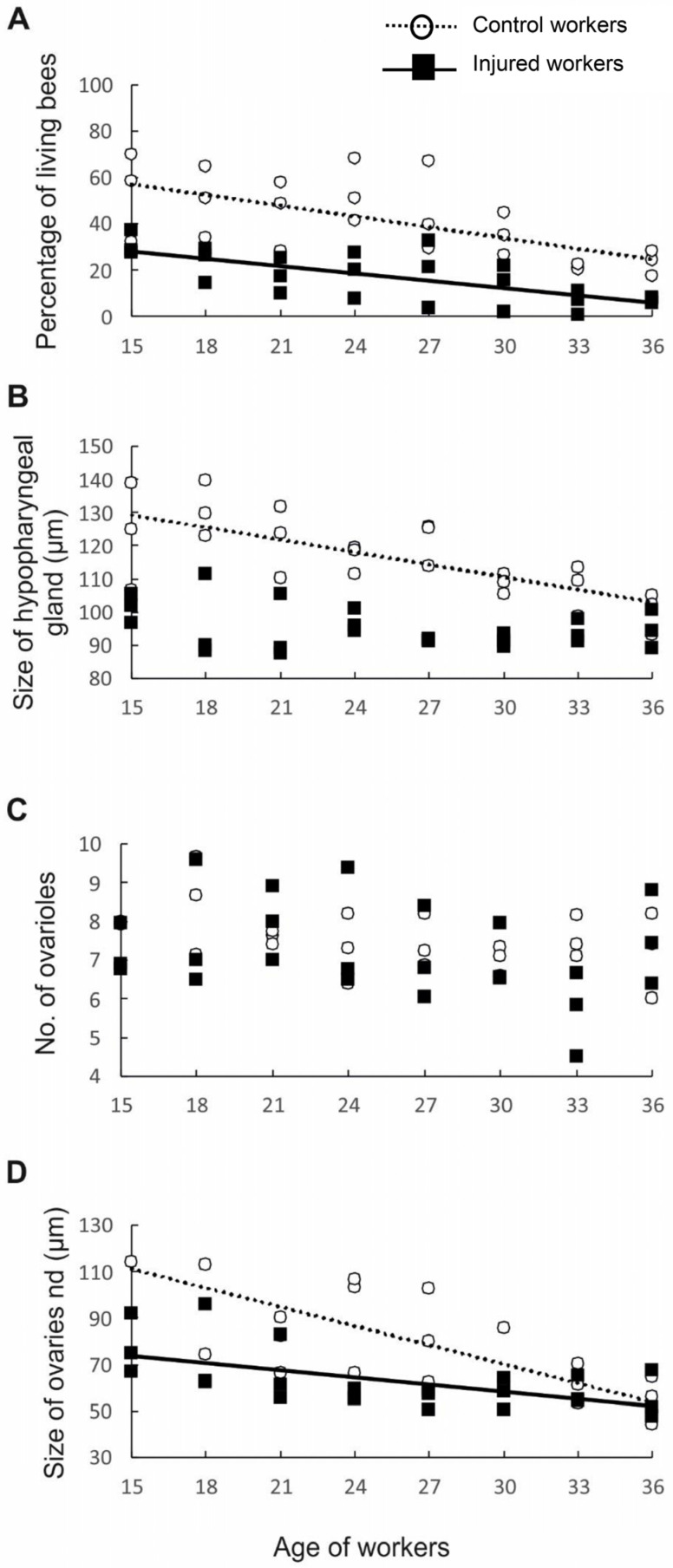
Survival and anatomical parameters of workers of different ages from two groups, namely, the control (hollow circles and dotted lines) and injured (black rectangles and full lines) groups: (**A**) survival of bees; (**B**) size of the hypopharyngeal gland; (**C**) number of ovarioles; and (**D**) ovariole size. The presence of lines indicates that the parameters are related to worker age.

## Data Availability

The data are in the Appendix A.

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
