# Peer review of "Reproductive Cessation and Post-Reproductive Lifespan in Honeybee Workers"

_biology, 2024, doi:10.3390/biology13050287_

Round 1
Reviewer 1 Report
Comments and Suggestions for Authors
Dear Authors,
You have done an original and interesting study.
I have a few comments.
1. Results. In Figure 1, the names of the graphs practically coincide with the names of the ordinate axis. It is enough to label the ordinate axes with units of measurement.
2. In the introduction and discussion there are few references to modern research: 4 for 2017-2018 and only 1 for the last 5 years. It would be appropriate to describe possible mechanisms of physiological aging of worker bees through injury. Most likely, this mechanism is associated with the activation of oxidative processes as a result of damage to the integument of insects. Quite a lot of articles on the connection between oxidative stress and aging and lifespan of insects have been published recently.
Author Response
Reviewer 1
You have done an original and interesting study.
I have a few comments.
- In Figure 1, the names of the graphs practically coincide with the names of the ordinate axis. It is enough to label the ordinate axes with units of measurement.
Thanks, added units (days) in the table
- In the introduction and discussion there are few references to modern research: 4 for 2017-2018 and only 1 for the last 5 years.It would be appropriate to describe possible mechanisms of physiological aging of worker bees through injury. Most likely, this mechanism is associated with the activation of oxidative processes as a result of damage to the integument of insects. Quite a lot of articles on the connection between oxidative stress and aging and lifespan of insects have been published recently.
Thanks, added this information in new version of our MS (line 164-172)
Reviewer 2 Report
Comments and Suggestions for Authors
The article titled " Reproductive Cessation and Post-Reproductive Lifespan in Honeybee Workers " provides valuable insights into the evolutionary enigma of post-reproductive lifespan, particularly focusing on its existence within honeybee societies. By demonstrating that older or injured worker bees invest fewer resources into their own reproduction compared to younger or uninjured ones, the research supports the kin selection explanation for the evolution of menopause. This finding contributes significantly to our understanding of relatedness and social cooperation in animal behavior.
However, there are a few points that require attention for improvement:
1. The figures should clearly indicate which groups represent the treatment and control conditions to ensure clarity in experimental design and results presentation.
2. The explanation regarding the lower reproductive results of injured worker bees lacks sufficient experimental evidence and literature support. Additional references are needed to strengthen this interpretation.
3. Providing photographs to visually illustrate the size of the hypopharyngeal gland and ovaries would enhance the clarity and impact of the results.
Overall, this study significantly advances our understanding of menopause evolution in honeybee societies. Addressing the points mentioned above would further strengthen the rigor and impact of the research.
Author Response
Reviever 2
The article titled " Reproductive Cessation and Post-Reproductive Lifespan in Honeybee Workers " provides valuable insights into the evolutionary enigma of post-reproductive lifespan, particularly focusing on its existence within honeybee societies. By demonstrating that older or injured worker bees invest fewer resources into their own reproduction compared to younger or uninjured ones, the research supports the kin selection explanation for the evolution of menopause. This finding contributes significantly to our understanding of relatedness and social cooperation in animal behavior.
However, there are a few points that require attention for improvement:
- The figures should clearly indicate which groups represent the treatment and control conditions to ensure clarity in experimental design and results presentation.
Thanks for this comment. The information about the markings the groups are in the description of figure 2. However, we added additionally graphical legend on the Figure 2.
- The explanation regarding the lower reproductive results of injured worker bees lacks sufficient experimental evidence and literature support. Additional references are needed to strengthen this interpretation.
The lower reproductive potential of injured workers is shown in results but also in the Figure 2 of MS. Additionally, some information about this (proportion of workers with different ovary activation) is in supplementary material. Unfortunately, there is no evidence in the literature about the changes in reproductive potential in the injured workers because we done it first time. There are evidence that injured workers life shorter life and are differ in behavior and physiology than control individuals but this is describe in our MS in line 84-88; 162-164; . We also add some information about oxidative processes which can be higher in injured individuals MS in line 164-172.
- Providing photographs to visually illustrate the size of the hypopharyngeal gland and ovaries would enhance the clarity and impact of the results.
Unfortunately, we have no photos from this experiment.
Overall, this study significantly advances our understanding of menopause evolution in honeybee societies. Addressing the points mentioned above would further strengthen the rigor and impact of the research.
Reviewer 3 Report
Comments and Suggestions for Authors
This paper examines the effect of age and injury on the size of hypopharyngeal glands and ovaries in hopelessly queenless worker honey bees. I find the data and manuscript compelling and well written.
The introduction should include more details of the age of workers that is most likely to have active ovaries when hopelessly queenless, currently the general trend is given. This will also speak to the suitability of the time period used in the experiment, for example, workers that emerge after the queen has been removed are not included and it may be that the absence of queen pheromone for that cohort would result in an even greater response than that found perhaps due to the effect of queen pheromone on ovaries, but potentially also in the quality or quantity of food fed to still-developing bees. Given the low proportion of older bees recaptured this would not be possible with a single timepoint. However it would be really cool to have the same ages sampled at a different point in time.
I may have missed it but I think it is worth summarising some information from the supplementary material about the proportion of workers with active ovaries in different colonies as there is quite a difference between colonies
Line 40-42, 212-214, 224-226: references
Author Response
Reviever 3
This paper examines the effect of age and injury on the size of hypopharyngeal glands and ovaries in hopelessly queenless worker honey bees. I find the data and manuscript compelling and well written.
The introduction should include more details of the age of workers that is most likely to have active ovaries when hopelessly queenless, currently the general trend is given. This will also speak to the suitability of the time period used in the experiment, for example, workers that emerge after the queen has been removed are not included and it may be that the absence of queen pheromone for that cohort would result in an even greater response than that found perhaps due to the effect of queen pheromone on ovaries, but potentially also in the quality or quantity of food fed to still-developing bees.
Thanks for this comment, we added some short information in line 123-124 in new version of MS. We also want to want to emphasize that the group of workers, collected as 15-days old individuals, spent whole adult life (after emerging) in queenless colony. Depside this injured individuals have less developed ovary compared to control workers (the number of ovarioles is similar but we didn't expect that these two groups will be differ in number of ovarioles in ovary). We also agree with the last sentence that workers, which develop after removal of queen can have better developed ovary in adult life. The individuals, which grow up in larval stage in orphaned colony develop into bees called "rebel workers" (e.g. Woyciechowski, M., & Kuszewska, K. (2012). Swarming generates rebel workers in honeybees. Current Biology, 22(8), 707-711.; Kuszewska, K., & Woyciechowski, M. (2015). Age at which larvae are orphaned determines their development into typical or rebel workers in the honeybee (Apis mellifera L.). PLoS One, 10(4), e0123404) but they are completely different physiologically and cannot be compared with normal or injured workers in this study (e.g. Strachecka, A., Migdał, P., Kuszewska, K., Skowronek, P., Grabowski, M., Paleolog, J., & Woyciechowski, M. (2021). Humoral and cellular defense mechanisms in rebel workers of Apis mellifera. Biology, 10(11), 1146.; Strachecka, A., Chobotow, J., Kuszewska, K., Olszewski, K., Skowronek, P., Bryś, M., ... & Woyciechowski, M. (2022). Morphology of Nasonov and Tergal Glands in Apis mellifera Rebels. Insects, 13(5), 401.). Moreover the diet in adults also can influence on ovary development (Hoover, S. E., Higo, H. A., & Winston, M. L. (2006). Worker honey bee ovary development: seasonal variation and the influence of larval and adult nutrition. Journal of Comparative Physiology B, 176, 55-63.) but in this study both experimental groups of bees were in the same condition during both larval and adult life.
Given the low proportion of older bees recaptured this would not be possible with a single timepoint. However it would be really cool to have the same ages sampled at a different point in time.
In this experiment we checked the changes in reproductive potential during the age (or more precisely physiological age) and for this reason we catch the bees with different age and also experimental groups but in the same timepoint. If we catch bees at different timepoint we will only learn how the environment affects reproductive potential, but not whether age matters. If we take the same ages bees but different point in.
I may have missed it but I think it is worth summarising some information from the supplementary material about the proportion of workers with active ovaries in different colonies as there is quite a difference between colonies
The most important information are attached in main text in MS. The supplementary information include data, which can be important for some readers but not for all. If we write and discuss this results in main text, it should bu included to main text. For our opinion this is only raw data and another presentation this data than in MS and for this reason, they are in only in Supplementary file and also for this reason we did't added this information in main text in MS.
Line 40-42, 212-214, 224-226: references
We added refererences in these lines (in new version of MS – there are line 55-57; 236-238; 248-250)